# Autoimmune Encephalitis with Antibodies: Anti-NMDAR, Anti-AMPAR, Anti-GQ1b, Anti-DPPX, Anti-CASPR2, Anti-LGI1, Anti-RI, Anti-Yo, Anti-Hu, Anti-CV2 and Anti-GABAAR, in the Course of Psychoses, Neoplastic Diseases, and Paraneoplastic Syndromes

**DOI:** 10.3390/diagnostics13152589

**Published:** 2023-08-03

**Authors:** Michał Braczkowski, Dariusz Soszyński, Alicja Sierakowska, Ryszard Braczkowski, Klaudia Kufel, Beata Łabuz-Roszak

**Affiliations:** 1Department of Physiology, Institute of Medical Sciences, University of Opole, 45040 Opole, Poland; 2Department of Human Physiology, Faculty of Medicine, Collegium Medicum in Bydgoszcz, Nicolaus Copernicus University in Torun, 87100 Torun, Poland; 3Student Scientific Society of Physiology, Department of Physiology, Institute of Medical Sciences, University of Opole, 45040 Opole, Poland; alicja.sierakowska@wp.pl (A.S.);; 4Student Association of Neurology, Department of Neurology, Institute of Medical Sciences, University of Opole, 45040 Opole, Poland; 5Specialist Hospital No. 1, 41900 Bytom, Poland; 6Department of Neurology, Institute of Medical Sciences, University of Opole, 45040 Opole, Poland; 7Department of Neurology, ST Jadwiga Regional Specialized Hospital, 45040 Opole, Poland

**Keywords:** autoimmune inflammation with antibodies, anti-NMDAR antibodies, anti-AMPAR antibodies, anti-GQ1b antibodies, anti-DPPX antibodies, anti-CASPR2 antibodies, anti-LGI1 antibodies, encephalitis, psychoses

## Abstract

Encephalitis is a condition with a variety of etiologies, clinical presentations, and degrees of severity. The causes of these disorders include both neuroinfections and autoimmune diseases in which host antibodies are pathologically directed against self-antigens. In autoimmune encephalitis, autoantibodies are expressed in the central nervous system. The incidence of this disease is approximately 4% of all reported cases of encephalitis. Autoimmune encephalitis can be induced by antibodies against neuronal surface antigens such as N-methyl-D-aspartate-activated glutamate receptors (NMDAR), α-amino-3-hydroxy-5-methyl-4-isoxazole propionate receptors (AMPAR) or gangliosides GQ1b, DPPX, CASPR2, LGI1, as well as by antibodies against neuronal intracellular antigens. The paper presents a number of both mental and neurological symptoms of autoimmune encephalitis. Moreover, the coexistence of psychoses, neoplastic diseases, and the methods of diagnosing autoimmune encephalitis are discussed. Attention was also drawn to the fact that early diagnosis, as well as early initiation of targeted treatment, increases the chance of a successful course of the therapeutic process. Strategy and Methodology: The articles on which the following paper was based were searched using search engines such as PubMed and Medline. Considering that anti-NMDAR antibodies were first described in 2007, the articles were from 2007 to 2023. The selection of papers was made by entering the phrases “autoimmune encephalitis and psychosis/paraneplastic syndromes or cancer”. The total number of articles that could be searched was 747, of which 100 items were selected, the most recent reports illustrating the presented topic. Thirty-four of them were rejected in connection with case reports or papers that could not be accessed.

## 1. Introduction

Encephalitis is a severe disorder with many possible causes with varying clinical presentations and degrees of severity. The causes of these disorders include both neuroinfections and autoimmune diseases [1,2]. There are reports indicating that various inflammatory conditions may outnumber infectious diseases in the central nervous system [3].

Autoimmune encephalitis is a heterogeneous group of inflammatory diseases of the central nervous system of various etiologies, requiring complex differential diagnosis. It is a severe clinical condition in which there is a misdirected immune response and directing autoantibodies against self-antigens, with expression taking place in the central nervous system [3,4]. Autoimmune encephalitis is a heterogeneous group of diseases with central nervous system inflammation, characterized by a wide range of symptoms, both neurological and psychiatric. They can occur at any age, and it is characteristic that they develop as a rapidly progressive encephalopathy (usually in less than 6 weeks) caused by encephalitis [5,6]. According to Zuliani et al. [7], autoimmune encephalitis describes several types of different diseases with different pathophysiology, the understanding of which is necessary for the selection of appropriate therapy. In turn, this understanding requires diagnosis based on the identification of autoantibodies associated with this neurological disorder.

Autoimmune encephalitis is not uncommon these days. Initially, the incidence of this disease was a small proportion of all reported cases of encephalitis. However, the last two decades have seen a significant increase in the incidence of autoimmune encephalitis. This is related to the development of diagnostic methods and the discovery over this time of numerous autoantibodies directed against the extracellular domains of neuroglial proteins. Advances in autoimmune encephalitis research have led to the identification of new biomarkers that have allowed for an extended diagnostic approach to these disorders. Thus, there was a significant increase in the number of reported cases of autoimmune encephalitis. Due to the multitude of symptoms of autoimmune encephalitis, this disease is still underestimated. This makes it impossible to unequivocally determine the percentage of morbidity among the general population. Recent epidemiological studies even suggest that autoimmune encephalitis is probably as common as infectious encephalitis, with a prevalence estimated at 13.7/100,000 [5,6,7].

The factor that makes it difficult to make a diagnosis of autoimmune encephalitis is the time from the onset of the first symptoms. In the majority of reported cases, the median time to CSF antibody response ranges from a few days to a few weeks, with a peak frequency around the third week. However, the literature on the subject describes chronic cases, even from one to five years. Therefore, diagnostics based on differential tests of autoantibodies in the patient’s blood have become important [8,9].

The paper presents a number of both mental and neurological symptoms of autoimmune encephalitis. Moreover, the coexistence of psychoses, neoplastic diseases, and the methods of diagnosing autoimmune encephalitis are discussed. Attention was also drawn to the fact that early diagnosis, as well as early initiation of targeted treatment, increase the chance of a successful course of the therapeutic process.

## 2. Antibodies in the Course of Autoimmune Encephalitis

One of the essential elements of the diagnosis of autoimmune encephalitis is the detection and identification of specific antineuronal antibodies. This disease can be induced by antibodies directed against neuronal surface antigens, such as glutamate receptors activated by N-methyl-D-aspartate (NMDAR), α-amino-3-hydroxy-5-methyl-4-isoxazole propionate receptors (AMPAR), and gangliosides GQ1b, DPPX, CASPR2, LGI1, as well as by antibodies directed against intracellular antigens of neurons. Identification of these antibodies determines the initiation of specific therapeutic actions [8,9,10].

Currently, numerous antibodies inducing the described autoimmune disease are distinguished. According to Wandinger et al. [10], newly identified forms of autoimmune encephalitis are often associated with antineuronal antibodies directed not only against intracellular antigens but, above all, against surface antigens. According to these authors, in as many as 63% of cases of diagnosed autoimmune encephalitis, we deal with antibodies against surface antigens; in 30% of cases, we deal with antibodies against intracellular antigens; and only in 7% of patients [Figure 1], we encounter both types of antibodies. Wandinger et al. [10] estimated the frequency of individual antibodies in their work.

As reported, with the highest frequency of 36%, a single anti-neuronal antibody directed against NMDAR antigens is identified. At the level of 9%, a single anti-neuronal antibody directed against AMPAR antigens is identified. At the level of 9%, a single antineuronal antibody directed against LGI1 antigens is identified. At the level of 9%, a single antineuronal antibody directed against CASPR2 antigens is identified. At the level of 8%, a single antineuronal antibody directed against Ri antigens is identified. At the level of 7%, a single antineuronal antibody directed against Yo antigens is identified. At the level of 5%, a single antineuronal antibody directed against Hu antigens is identified. At the level of 4%, a single anti-neuronal antibody directed against CV2 antigens is identified. At the level of 3%, a single antineuronal antibody directed against GABA_B_R antigens is identified. At the 2% level, a single antineuronal antibody directed against the Ma2/Ta antigens is identified. At the 1% level, a single anti-neuronal antibody directed against PCA-2 antigens is identified. At the 1% level, a single anti-neuronal antibody directed against amphiphysin antigens is identified. These authors, in their work, also determined the frequency of coexistence of individual antineuronal antibodies in the course of autoimmune encephalitis at the level of 3% for antibodies directed against NMDAR and CASPR2 antigens, 2% for antibodies directed against NMDAR and GABA_B_R antigens, 1% for antibodies directed against antigens NMDAR and Hu, and 1% for antibodies directed against Hu and CV2 antigens [8,9,10,11,12] [Figure 2].

Due to the frequency of occurrence, particular attention should be paid to antibodies against glutamate receptors induced by N-methyl-D-aspartate (NMDAR), antibodies to α-amino-3-hydroxy-5-methyl-4-isoxazole propionate (AMPAR) receptors, and antibodies against gangliosides CASPR2, LGI1, as well as recently described GQ1b and DPPX [8,9].

### 2.1. Autoimmune Encephalitis with Anti-NMDAR Antibodies

Autoimmune encephalitis caused by antibodies directed against the NMDA (N-methyl-D-aspartate) receptor subtype was first described in 2007. It has since entered the mainstream of interest in neurology and psychiatry. Some patients with rapidly progressive psychiatric symptoms or cognitive impairment, seizures, abnormal movements, or coma of unknown cause have been observed to have a previously undiagnosed autoimmune disease.

Anti-NMDAR encephalitis is the most common form of autoimmune antibody-associated encephalitis. The prevalence of the disease was initially estimated at approximately 4% of all reported cases of encephalitis. In the study, the aim of which was to determine the cause of the first psychotic episode, in less than 4% of participants, the background was defined as non-psychiatric, of which anti-NMDAR antibodies were detected in as many as 75% [11,12]. The condition can affect people of all ages, but the highest incidence rates are in children and young adults. More than 37% of patients are under the age of 18 at the onset of the disease. The disease affects women more often than men. As many as 80% of patients are women, with an average age of onset of 21 years [12]. In more than half of cases, the disease is associated with cancer and responds to treatment but may recur. Ovarian teratoma was observed in 58% of women over 18 years of age. In addition to age, the presence of the tumor also depends on gender and ethnicity, being slightly more common in black women than in white women. Patients treated with tumor resection and immunotherapy (corticosteroids, intravenous immunoglobulins, or plasma exchange) respond more quickly to treatment and are less likely to require second-line immunotherapy (cyclophosphamide or rituximab or both) than patients without a tumor who receive similar initial immunotherapy. In over 75% of all patients, a significant recovery occurs, which occurs in the reverse order of the development of symptoms and is associated with a decrease in antibody titer [13].

In this disease, autoantibodies serve as a diagnostic marker and alter NMDAR-related synaptic transmission. N-methyl-d-aspartate receptor antibodies belong to the immunoglobulin G1 subclass and are capable of activating a complement on human embryonic kidney cells expressing the N-methyl-d-aspartate receptor. Patients’ antibodies cause a titer-dependent, reversible decrease in synaptic NMDAR through a mechanism of cross-linking and internalization. Based on models of pharmacological or genetic disruption of NMDARs, these antibody effects reveal a probable pathogenic link between receptor depletion and the clinical features of anti-NMDAR encephalitis [10]. The pathogenesis of autoimmune encephalitis with anti-NMDAR antibodies most likely has its origin in neoplastic lesions as well as in virus-infected neurons. They release antigens for NMDA receptors (NMDARs), which are taken up by antigen-presenting cells (APCs) after apoptosis. APCs migrate to local lymph nodes, where, as a result of cooperation with CD4+ T lymphocytes, naive B lymphocytes differentiate into plasma cells and memory cells. The anti-NMDAR antibodies produced by these plasma cells are transported with the blood to the regions of the brain and synaptic connections, where they compete with the GluN1 and RphB2 subunits and lead to neurotransmitter dysregulation. Thus, they become the cause of psychotic symptoms [13].

At the onset of symptoms, distinguishing autoimmune encephalitis from a primary psychiatric disorder is difficult. Autoimmune encephalitis with anti-NMDAR antibodies may begin with non-specific flu-like symptoms followed by fairly rapid (within days to six weeks) development of neuropsychiatric symptoms. The disease presents as a rapidly progressing encephalopathy with acute or subacute onset. The main symptom, which occurs in 90% of patients, is psychiatric disorders, usually of acute onset. The coexistence of a number of symptoms is significant, which is not characteristic of any mental illness. Most patients with anti-NMDAR encephalitis develop a multi-stage disease that progresses from psychosis and seizures to an unresponsive state with catatonic features often associated with autonomic and respiratory instability. In the cognitive sphere, abnormalities such as disorientation, confabulation, confusion, or memory deficits can be observed. The clinical picture is complemented by pathology noted in the motor sphere. Symptoms such as facial and tongue dyskinesia, chorea, stereotypia, athetosis, and dystonia, as well as epileptic seizures (mainly tonic-clonic morphology, focal seizures, and absence seizures), are not uncommon. The severity of symptoms often requires intensive care. Other than clinical evaluation, there are no specific prognostic biomarkers. The disease is characterized by an acute and aggressive course; however, in the case of correct diagnosis and promptly implemented treatment, most patients achieve permanent remission. Correctly defined criteria of diagnosis determine the period of occurrence of symptoms as not longer than 3 months. In addition to obtaining a positive result of anti-NMDAR antibodies, additional tests such as EEG or MRI should be performed to confirm symptoms associated with autoimmune encephalitis with anti-NMDAR antibodies, such as EEG or MRI [1,8,14,15,16,17].

### 2.2. Autoimmune Encephalitis with Anti-AMPAR Antibodies

Autoimmune encephalitis can be induced by antibodies directed against α-amino-3-hydroxy-5-methyl-4-isoxazole propionate receptors (AMPAR), which are neuronal surface antigens. The disease is a rare type of autoimmune encephalitis, which was first described by Lai et al. [18]. α-Amino-3-hydroxy-5-methyl-4-isoxazole propionate receptors are expressed throughout the central nervous system, particularly in the hippocampus and other limbic regions. Anti-AMPAR antibodies are directed against extracellular epitopes of the GluA1 or GluA2 subunits of this receptor. Anti-AMPAR encephalitis is most common in middle-aged women, and most patients have an acute or subacute onset. As with anti-NMDAR autoimmune encephalitis, suspected anti-AMPAR autoimmune encephalitis often shows inflammatory changes in the cerebrospinal fluid in laboratory tests. Therefore, it can be expected that in patients with anti-NMDAR and anti-AMPAR encephalitis, tests will show inflammatory changes in the cerebrospinal fluid in most cases. This is important in the diagnosis of this autoimmune disease, as it allows for making appropriate and quick therapeutic decisions. Encephalitis caused by anti-AMPAR antibodies is a neurological disease that is often accompanied by the presence of tumors and, therefore, may cover the entire spectrum of the paraneoplastic syndrome [8,18].

The thymus is the central organ of the immune system, and it is believed that tumors of the thymus may be the initiators of many neurological disorders. Recently, there has been increasing evidence that thymomas are associated with autoimmune encephalitis. Omi et al. [19] were the first to describe a case of encephalitis directed against the α-amino-3-hydroxy-5-methyl-4-isoxazole propionic acid receptor, whose clinical recurrence coincided with thymoma recurrence. Clinical recurrence of anti-AMPAR encephalitis after recurrence of the initially detected tumor has not been previously reported. The patient had been in remission for 34 months and showed clinical relapse 3 months after the thymoma recurrence was detected. This case highlights the pathogenic importance of specific tumor antigens as a trigger for anti-AMPAR antibody production and disease induction. In turn, Song et al. [20] tracked data from 1 January 2011 to 1 October 2021 from PubMed, Web of Science, Ovid, and CNKI platforms to analyze general demographic characteristics, frequency of symptoms and associations, and prognosis results in the treatment of 68 patients with naty-AMPAR autoimmune encephalitis in combination with thymoma. Clinical manifestations were mainly cognitive changes (70.6%), psychiatric disorders (57.4%), and epilepsy (50.0%). Treatment included immunotherapy and thymoma therapy, with 79.7% of patients improving post-treatment. In comparison, 20.3% of patients had a poor prognosis. Relapse occurred in 14.8% of patients. According to these authors, thymoma-associated autoimmune encephalitis is a disease entity that absolutely requires long-term follow-up of chest CT results. Zhang et al. [21] report that inflammation of the limbic system of the brain is very often accompanied by antibodies against AMPAR receptors. Autoimmune encephalitis in this form is clinically characterized by a subacute disorder consisting of short-term memory loss, confusion, abnormal behavior, and seizures. According to these authors, most patients with anti-AMPAR encephalitis had malignant complications such as thymoma, small cell lung cancer, breast cancer, and ovarian cancer. Most patients with anti-AMPAR encephalitis showed a partial neurological response to immunotherapy. McCombe et al. [22] confirmed the occurrence of thymoma as the most common tumor accompanying autoimmune encephalitis with anti-AMPAR antibodies. In addition, these authors report that in anti-AMPAR encephalitis, the concomitant antibodies predict a different clinical picture than encephalitis and thymoma. In isolated cases, anti-AMPAR autoimmune encephalitis is accompanied by unexpected disease entities. Despite not fully understanding the connections between its co-diagnosis with thymoma and myasthenia gravis, Li et al. [23] were the first to describe the autoimmune profile and clinical picture of a patient with α-amino-3-hydroxy-5-methyl-4-isoxazole propionic acid and myasthenia gravis. This points to the extraordinary complexity of thymoma-associated autoimmune networks. Joubert et al. [24] report that the clinical picture of autoimmune encephalitis with anti-AMPAR antibodies is variable. Most patients present with symptoms suggestive of autoimmune limbic encephalitis. They may also show no symptoms or may manifest as severe encephalitis that evolves to a state of minimal consciousness and diffuse cerebral atrophy. Contrary to other reports, in the group of seven patients analyzed, these authors observed only one patient with epileptic seizures and only two patients with cancer—one with lung cancer and one with thymus cancer. These authors clearly emphasize that due to poorly understood clinical features, patients with suspected autoimmune encephalitis associated with the presence of antibodies to the α-amino-3-hydroxy-5-methyl-4-isoxazole propionic acid receptor should be screened for the presence of anti-AMPR antibodies both in the serum and in the cerebrospinal fluid [18,19,20,21,22,23,24,25].

### 2.3. Autoimmune Encephalitis with Anti-GQ1b, Anti-DPPX, Anti-CASPR2, and Anti-LGI1 Antibodies

Autoimmune encephalitis can also be induced by antibodies directed against gangliosides GQ1b, DPPX, CASPR2, and LGI1 [8,26,27,28,29,30].

In the 1950s, Bickerstaff and Fisher independently reported cases with the unique symptom of ophthalmoplegia and ataxia. Neurological symptoms were usually preceded by a previous infection, and most patients recovered spontaneously. However, both authors saw some similarities with Guillain–Barré syndrome, such as the presence of peripheral neuropathy and albuminocytological dissociation of the cerebrospinal fluid. The discovery of anti-GQ1b antibodies in patients with Fisher’s syndrome and later Bickerstaff’s encephalitis was crucial in providing the necessary evidence to conclude that both conditions were, in fact, part of the same spectrum of diseases due to their common clinical and immunological characteristics. The occurrence of anti-GQ-1b antibodies is associated with a variety of disorders. According to Yoshikawa et al. [27], confirmation of the presence of anti-GQ1b antibodies may affect vision disorders. Only recently have these antibodies been associated with autoimmune encephalitis [26,27].

Autoimmune encephalitis caused by autoantibodies directed against the dipeptidyl peptidase-like protein (DPPX), a potassium channel subunit, is rare. In autoimmune encephalitis with antibodies against DPPX receptors, the cerebrospinal fluid is often characterized by inflammation. The disease is typically characterized by a triad of symptoms that include weight loss, central nervous system hyperactivity, and cognitive deficits. However, recent reports suggest that the clinical picture may be more heterogeneous. Although blood and cerebrospinal fluid analyses often show normal cell counts, high titers of DPPX antibodies in the blood and cerebrospinal fluid confirm autoimmune encephalitis with anti-DPPX antibodies. Piepgras et al. [28] confirmed the pathogenic role of anti-DPPX antibodies in anti-DPPX encephalitis in studies on the hyperexcitability of neurons of the enteric nervous system and the reduction of the level of DPPX from the membranes of hippocampal neurons. Xiao et al. [29] conducted a retrospective analysis of nine patients with anti-DPPX encephalitis to expand the clinical and imaging phenotypes of anti-DPPX encephalitis. Nine patients (median age 51 years; range 14–65 years) were identified with symptoms of prodromal fever, diarrhea, or weight loss, followed by rapidly progressive encephalopathy characterized by cognitive impairment. They described three patients with anti-DPPX encephalitis who had sleep disturbance with abnormal sleep behavior with rapid eye movements, limb paralysis, and severe pleocytosis. One patient who received methylprednisolone therapy and a trial of tacrolimus showed significant improvement and had no recurrence during the 6-month follow-up. However, these authors emphasize that further research is warranted to explain the entire clinical spectrum of anti-DPPX encephalitis, its pathogenic mechanisms, and prognosis in the case of long-term immunosuppressive therapy [28,29,30].

CASPR2 and LGI1 are proteins that are part of a complex of proteins related to the voltage-gated potassium channel VGKC (VGKC). Autoantibodies against extracellular proteins LGI1 and Caspr2 were first described in 2010. This has significantly changed the view on the clinical relevance of antibodies associated with the voltage-gated potassium channel. Autoimmune encephalitis with autoantibodies to inactivated leucine-rich glioma 1 (LGI1) accounts for the diagnosis of anti-LGI1 autoimmune encephalitis. In turn, the diagnosis of antibodies directed against contacttin-like protein 2 (CASPR2) is responsible for anti-CASPR2 encephalitis. Encephalitis associated with antibodies to CASPR2 is a rare autoimmune disease. The main clinical manifestations are believed to be convulsions, memory loss, psychiatric symptoms, dizziness, and sleep disturbances. The most common clinical symptoms in both of these subtypes are epileptic seizures, memory deficits, mental disorders, and disturbances of consciousness. Currently, all of these antibodies are classified as VGKC complex antibodies and are generally considered to be of similar clinical value. However, research over the last few years shows that the immune responses mediated by these antibodies have different clinical relevance. Antibodies against LGI1 and Caspr2 are associated with different but well-defined neurological syndromes. Both share common and distinct clinical features mediated by autoantibodies directed against different proteins complexed with voltage-gated potassium channels.

Limbic encephalitis is an etiologically heterogeneous disease because it can occur both as a paraneoplastic disease and in patients without oncological disease. Various neuronal autoantibodies have been found in patients with this type of encephalitis. According to previous studies, antibodies against the VGKC complex are most often present, and among the proteins included in this complex, antibodies against the LGI1 protein, a protein that plays an important role in synaptic transmission by participating in the formation of pre- and post-synaptic protein complexes.

Anti-CASPR2 and anti-LGI1 encephalitis is the most common subtype in the elderly, with the majority of CSF findings normal. As a result, in patients with encephalitis with anti-NMDAR, anti-AMPAR, and anti-DPPX antibodies, in most cases, the tests will show inflammatory changes in the cerebrospinal fluid, supporting the diagnosis of an autoimmune disease. Unfortunately, this will not be the case for most patients with anti-LGI1 and anti-CASPR2 antibodies. However, in the case of suspected autoimmune encephalitis with anti-LGI1 and anti-CASPR2 antibodies in the elderly, the results of these tests should not lead to a decision to abandon the serum antineuronal antibody test [31,32,33,34].

## 3. Autoimmune Encephalitis and Psychosis

Schizophrenia is a disease resulting from functional and structural changes in the brain. Psychopathological symptoms indicating a diagnosis of schizophrenia include disturbances in the course and content of thinking. They are characterized by the occurrence of delusions: of reference, of persecution, and of influence. Commenting voices or echoes of thought often co-occur as well. Another characteristic symptoms of the disease are negative symptoms. The risk of occurrence in the general population is 1–1.5% [1,2]. However, single psychotic episodes (psychotic-like experience—PLE) may affect as many as 5–8% of the population [3].

Psychosis is a mental disorder in which the patient experiences disturbances in perceiving reality, and sometimes also consciousness, as a result of which he is unable to maintain a critical attitude towards realism. Psychosis can manifest itself in primary psychotic disorders, neurological diseases, and disease states [4]. Thus, the positive symptoms presented by the patient are not always the first manifestation of schizophrenia. Autoimmune encephalitis is an example of a disease in which psychotic symptoms are not uncommon. The first-time symptoms of autoimmune encephalitis with anti-NMDAR antibodies include delirium, amnesia, and epileptic seizures. Symptoms from the spectrum of psychopathology are also characteristic, such as mood disorders, behavioral disorders, psychosis, or catatonia.

Psychosis is a serious mental disorder that is characterized by frequent delusions and hallucinations. Its special feature is defective or completely lost contact with reality, which is associated with a completely different interpretation and perception of reality than in the case of healthy people. Loss of contact with reality is manifested in psychosis by hallucinations and delusions, as well as disorganization in the way of expressing emotions, behavior, thinking, and communicating. In the course of autoimmune encephalitis with anti-NMDAR antibodies, the mental status of the patient changes depending on the severity of the inflammatory changes and the duration of the disorder. In the initial period, only non-specific prodromal symptoms are observed, indicating the onset of the disease before specific symptoms occur, which appear as early as 1–2 weeks after diagnosis and intensification of inflammation. Emerging psychosis is characterized by a number of psychiatric symptoms, among which the most common are delusions, hallucinations, mania, anxiety, speech disorders, catatonia, abulia, anhedonia, insomnia, and frequent epileptic seizures. Neurological complications, movement disorders, dysautonomia (fluctuations in blood pressure and heart rate), hypoventilation, and seizures may occur simultaneously. These symptoms prove that emerging psychosis does not have to be a symptom of schizophrenia, depression, dementia, bipolar affective disorder, or addiction to psychoactive substances and may be the result of autoimmune encephalitis [35,36].

## 4. Relationship between Autoimmune Encephalitis and Neoplastic Diseases and Paraneoplastic Syndromes

Malignancies, usually ovarian teratoma and herpetic encephalitis, are known triggers of NMDAR autoimmunity. The humoral response, which is the basis of the pathophysiology of autoimmune encephalitis, leading to the production of antineuronal antibodies by plasma cells directed against their surface antigens, e.g., NMDAR, is important not only in the diagnosis of these neurological disorders but also allows to estimate the risk of cancer. Thus, anti-neuronal antibodies make it possible to assess the risk of cancer in the course of autoimmune encephalitis, and some of them have a significant relationship with specific cancers. The presence of specific antibodies in autoimmune encephalitis may indicate a specific type of tumor and may lead to improvement or stabilization of symptoms related to its treatment. On the other hand, in the case of the paraneoplastic nature of the disease, diagnosis based on antineuronal antibodies facilitates the targeting of the cancer search.

Clinically, paraneoplastic encephalitis may not be distinct from non-paraneoplastic encephalitis. For this reason, when a patient is diagnosed with autoimmune encephalitis, targeted oncology diagnostics should be performed. Autoimmune encephalitis may be paraneoplastic or, in a small number of cases, be unrelated to cancer. In adults, the presence of an antibody against anti-Hu intracellular antigens is almost always associated with an existing tumor. In addition, anti-neuronal antibodies with a high risk of cancer include anti-Ri, anti-Yo, anti-Ma2/Ta, anti-CV2, and anti-amphiphysin. According to Gultekin et al. [37], the most common types of cancer associated with the course of autoimmune encephalitis include:Lung cancer, with an incidence of 50% of patients with autoimmune encephalitis, of which 40% are small cell lung cancer, 10% are non-small cell lung cancer;Testicular tumors, with an incidence of 20% of patients with autoimmune encephalitis;Breast cancer, with an incidence of 8% of patients with autoimmune encephalitis;Ovarian teratoma, with an incidence of 4% of patients with autoimmune encephalitis;Hodgkin’s disease, with an incidence of 4% of patients with autoimmune encephalitis;Thymus, with an incidence of 2% of patients with autoimmune encephalitis;Other cancers, with an incidence of 8% of patients with autoimmune encephalitis.

According to the same authors, only 4% of patients who are found to have any anti-neuronal antibodies, thus confirming autoimmune encephalitis, are not affected by any type of cancer.

Antibodies directed against intracellular antigens, apart from the targeted diagnosis of autoimmune encephalitis, also allow us to estimate the occurrence of classic paraneoplastic disorders. In turn, other most common antibodies against surface antigens, such as anti-LGI1 or anti-CASPR2, are associated with typical non-paraneoplastic forms of limbic encephalitis [38,39,40,41,42,43,44,45,46,47,48,49,50].

Also worth mentioning is the encephalopathy associated with Hashimoto’s disease. This is a controversial and poorly understood disease entity with a still unclear etiology. The incidence is estimated at 2.1/100,000, with a significant predisposition among adults between the fourth and sixth decades of life, with a definite predominance among women (4:1) than men [51]. The substrate of the disease is most likely based on autoimmune processes. Basic laboratory tests usually show no significant abnormalities. TSH levels are within normal limits, and only about 30% of patients have hypothyroidism. Nevertheless, elevated serum anti-TPO and/or anti-TGAb levels are noted in every patient [52,53]. Moreover, in about 62–75% of patients, antibodies characteristic of autoimmune encephalitis, such as anti-NMDAR, anti-GABAAR, anti-Caspr2, anti-LGi1, or anti-AMPAR, are found in the cerebrospinal fluid [54,55].

## 5. Diagnostics of Autoimmune Encephalitis in Adults and Children

According to the latest recommendations from 2021 [56], when autoimmune encephalitis is clinically suspected, the procedure begins with brain imaging and cerebrospinal fluid (CSF) analysis. One of the key elements of diagnostics is testing for the presence of antineuronal antibodies, not only in the cerebrospinal fluid but, above all, in the blood serum. Although there is a marked intrathecal synthesis of anti-N-methyl-d-aspartate receptor antibodies, absolute levels of anti-N-methyl-d-aspartate receptor antibodies are generally higher in serum than in cerebrospinal fluid.

However, the diagnostic procedure is different in children and adults.

The diagnostic algorithm for autoimmune encephalitis in adults first involves confirming focal or multifocal brain pathology suggestive of autoimmune encephalitis [56]. For this purpose, an MRI examination with or without contrast is performed. If the MRI result is negative, or if the patient has encephalopathy or frequent seizures, an EEG is then performed. If the result of the MRI test is negative and the diagnosis after the initial tests is still unclear, the FDG-PET test is performed in turn. In the next stage of the diagnostic procedure, the inflammatory etiology should be confirmed and other diagnoses excluded. For this purpose, we perform a CSF test, including anti-neuronal antibody testing, and a blood test that includes anti-neuronal antibody testing. If the diagnosis is still unclear, a brain biopsy is needed. The final step in the diagnostic procedure is cancer screening. For this purpose, we perform a CT scan of the chest, abdomen, and pelvis, mammography/MRI of the breast, and ultrasound examination of the pelvis or testicles. In the event of negative results, we perform the FDG-PET test.

Pediatric encephalitis should be diagnosed based on the clinical picture and additional tests, including anti-neuronal antibodies [57,58]. Thus, the diagnostic algorithm for autoimmune encephalitis in children first includes the verification of the results of basic tests, imaging, and anti-neuronal antibodies in order to determine their relationship with autoimmune encephalitis. In the case of negative results, there is a need to consider other causes of symptoms in a pediatric patient with a primary suspicion of symptoms of autoimmune encephalitis. In case of positive results, it should be verified whether the present antibodies are related to pediatric autoimmune encephalitis: anti-NMDAR, anti-MOG, anti-GAD65, anti-GABAA/BR, anti-GlyR, anti-m-GluR5, and anti-dopamine receptor. Negative results are likely to be non-antibody pediatric encephalitis. If the result is positive, pediatric antibody-associated encephalitis is certain.

In the diagnosis of anti-neuronal antibodies when autoimmune encephalitis is suspected, the gold standard is an indirect immunofluorescence test (IIFT) or an immunohistochemical method using animal tissues as a substrate and then performing highly specific tests to confirm—immunoblot for most antibodies against intracellular antigens or cellular tests, so-called cell-based assay (CBA), for antibodies directed against surface or synaptic antigens. Currently, the CBA test, which is a modification of the indirect immunofluorescence method, is quite commonly used. In this assay, cells transfected with selected target antigens for anti-neuronal antibodies are used as a substrate. The positive reaction is the characteristic fluorescence of cells transfected with primary specific human autoantibodies and fluorescein-labeled secondary anti-human antibodies, which allows differentiation from non-specific reactions and obtaining highly reliable results.

In the study and differentiation of the pathophysiology of autoimmune encephalitis, comprehensive panel studies of antineuronal antibodies are recommended, taking into account the importance of antineuronal antibodies of the IgG class only. This diagnosis is based on qualitative indirect immunofluorescence (IFA) tests for the detection of anti-NMDAR IgG. High levels of these antibodies correlate with disease severity and variable response to treatment. The study of antineuronal autoantibodies in suspected autoimmune encephalitis is of unquestionable importance. It allows us to identify the subtype of autoimmune encephalitis, which is a prognostic factor regarding the prognosis and success of therapy, enabling its monitoring. Currently, it is recommended to perform antineuronal antibody tests with a high clinical suspicion of autoimmune encephalitis, even in the case of normal results of the basic CSF test. According to the recommendations, the sensitivity and specificity of the analysis in serum or cerebrospinal fluid differ depending on the tested antineuronal antibodies; therefore, it is recommended to perform the test in both samples [59]: “In the case of suspected autoimmune or paraneoplastic encephalitis associated with the presence of antibodies on the surface of neurons, CSF screening should be mandatory” [1,60,61,62,63,64,65].

## 6. Summary

In recent years, autoimmune encephalitis is increasingly recognized as one of the causes of epilepsy, especially drug-resistant epilepsy. The cause of this disorder is autoantibodies directed against antigens, which are ion channels, receptors, and other synaptic proteins involved in neuronal transmission and brain plasticity, sometimes also in cytotoxic processes. Autoimmune encephalitis with the antibodies listed in the article can manifest itself in many ways. It is not uncommon for many different symptoms, both neurological and psychiatric, to occur. The clinical picture of autoimmune encephalitis with the presence of antibodies varies depending on their subtype and individual factors.

In the last decade, significant progress has been made in the diagnosis of autoimmune encephalitis. With effective differential diagnosis, levels of evidence of autoimmune encephalitis (possible, probable, or certain) with a specific antibody subtype are obtained, which can lead to the implementation of rapid immunotherapy. Early diagnosis and early initiation of targeted treatment increase the chance of a successful therapeutic process and may help patients achieve a good prognosis. However, in order to improve the diagnosis and treatment of autoimmune encephalitis, further research is needed to develop more sensitive diagnostic tools, identify prognostic biomarkers and implement new treatments that significantly accelerate recovery.

## Figures and Tables

**Figure 1 diagnostics-13-02589-f001:**
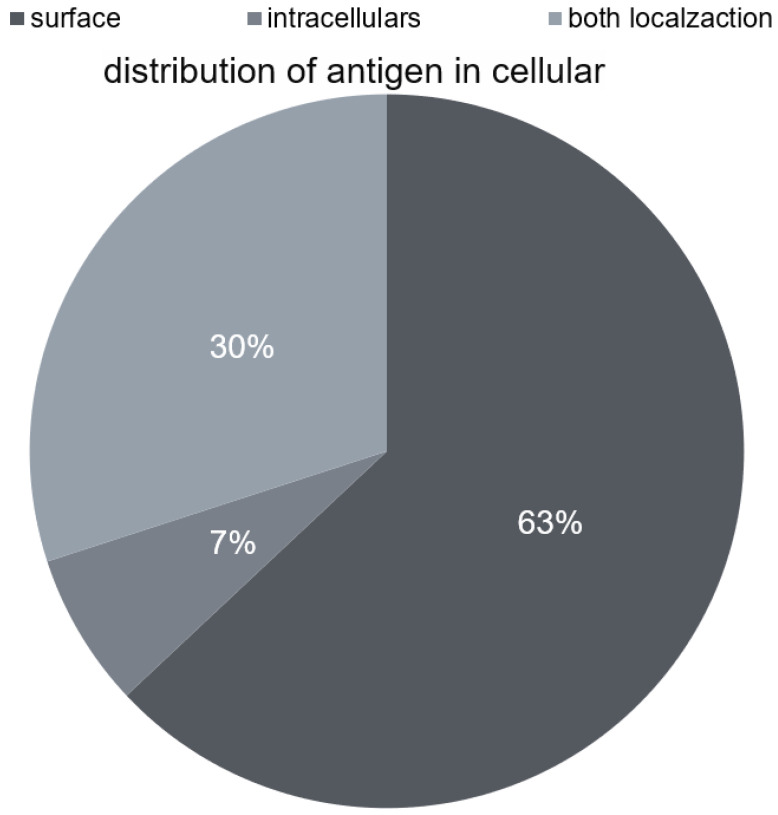
Distribution of antigen in cellular.

**Figure 2 diagnostics-13-02589-f002:**
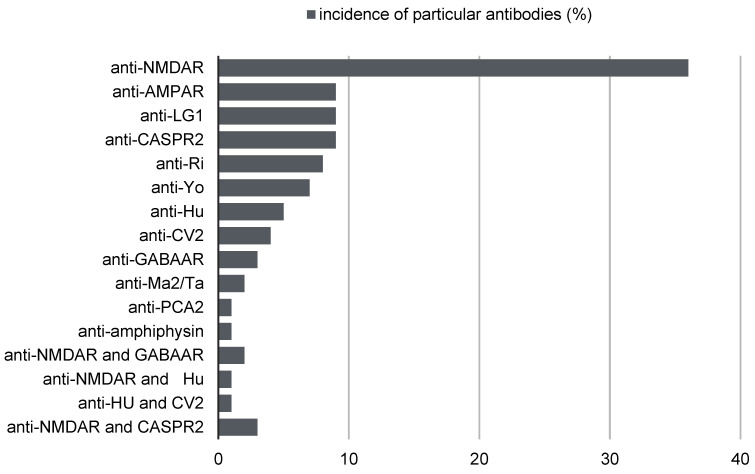
Incidence of particular antibodies.

## Data Availability

Not applicable.

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
