# Peer review of "Autoimmune Encephalitis with Antibodies: Anti-NMDAR, Anti-AMPAR, Anti-GQ1b, Anti-DPPX, Anti-CASPR2, Anti-LGI1, Anti-RI, Anti-Yo, Anti-Hu, Anti-CV2 and Anti-GABAAR, in the Course of Psychoses, Neoplastic Diseases, and Paraneoplastic Syndromes"

_diagnostics, 2023, doi:10.3390/diagnostics13152589_

Round 1
Reviewer 1 Report
The paper presented to me for review is a review article on autoimmune encephalitis with selected antibodies in the course of psychoses, neoplastic diseases and paraneoplastic syndromes. The paper is written quite well and exhausts the topic sufficiently.
However, before accepting the paper for publication, I would suggest some corrections and additions that could enhance its value:
1. there is a missing chapter on the search strategy and methodology on how the authors selected the articles in question, from what database and quantity they chose, what they rejected
2. the work would benefit from clarity if the authors included some of the information in tables or diagrams. it is usually very difficult to get information from continuous text and the reader quickly becomes discouraged
3. in the discussion, it is worth mentioning the most likely underdiagnosed disease entity of Hashimoto's encephalopathy, in which the antibodies mentioned by the authors may be present on the basis of: PMCID: PMC9496753
Author Response
in response
ad 1 "there is a missing chapter on the search strategy and methodology on how the authors selected the articles in question, from what database and quantity they chose, what they rejected" we added subsection with the title strategy and methodology;
ad 2 "the work would benefit from clarity if the authors included some of the information in tables or diagrams. it is usually very difficult to get information from continuous text and the reader quickly becomes discouraged" we added diagram "distribution of antigen in cellular" and graph "incidence of particular antibodies";
ad 3 "in the discussion, it is worth mentioning the most likely underdiagnosed disease entity of Hashimoto's encephalopathy, in which the antibodies mentioned by the authors may be present on the basis of: PMCID: PMC9496753" Chapter 4 has been expanded to include information on EH in the aspect of autoimmune encephalitis antibodies.

Reviewer 2 Report
Although this review article foucses on the high-frequency antibodies in autoimmune encephalitis, I would suggest a full list of autoimmune encephalitis-associated antibodies presenting in this review.
English language polishing may be benificial to this review article.
Author Response
Ad "Although this review article foucses on the high-frequency antibodies in autoimmune encephalitis, I would suggest a full list of autoimmune encephalitis-associated antibodies presenting in this review" In the title as well as the text, all antibodies associated with autoimmune encephalitis are listed

Round 2
Reviewer 1 Report
The authors fully addressed my suggestions and included everything in the manuscript. The paper is ready for publication.